# Midpalatal Suture Maturation Method for the Assessment of Maturation before Maxillary Expansion: A Systematic Review

**DOI:** 10.3390/diagnostics12112774

**Published:** 2022-11-13

**Authors:** Anis Shayani, Paulo Sandoval Vidal, Ivonne Garay Carrasco, Marco Merino Gerlach

**Affiliations:** 1Master Program in Dental Science, Faculty of Dentistry, Universidad de la Frontera, Temuco 4780000, Chile; 2School of Dentistry, Faculty of Medicine, Universidad Austral de Chile, Valdivia 5090000, Chile; 3Department of Pediatric Dentistry and Orthodontics, Faculty of Dentistry, Universidad de La Frontera, Temuco 4780000, Chile; 4Independent Researcher, Temuco 4780000, Chile; 5Independent Researcher, Valdivia 5090000, Chile

**Keywords:** midpalatal suture maturation method, maxillary expansion, cranial sutures, ossification, interdigitation

## Abstract

Assessment of midpalatal suture maturation is crucial before deciding which type of maxillary expansion technique will be performed to treat transverse discrepancies. In 2013, Angelieri et al. proposed a new method to evaluate midpalatal maturation using cone-beam computed tomography. The aim of this study was to systematically identify, evaluate, and provide a synthesis of the existing literature about this new method and to rigorously assess the methodological quality of these articles. A bibliographic search was carried out using PubMed, Cochrane Library, SciELO, LILACS, Web of Science, and Scopus using the terms midpalatal suture, cranial sutures, palate, maturation, interdigitation, ossification, maxillary expansion, evaluation, assessment, and assess. Quality assessment was performed using the Observational Cohort and Cross-Sectional Studies tool developed by the National Heart, Lung, and Blood Institute. Hence, 56 articles were obtained, of which only 10 met the selection criteria. We could not include any of the data into an analysis because of the large variation of the data collected and high methodological heterogeneity found among studies. Of all the studies included, 10% had poor quality, 70% fair, and 20% good quality, respectively. Even though age and sex play a role in midpalatal suture obliteration, there is a poor correlation between these variables. Thus, every patient should be assessed individually before choosing the best protocol for maxillary expansion. The midpalatal suture maturation method has the potential to be used for diagnostic purposes, but clinicians should be cautious of routinely using it because an extensive training and calibration program should be performed prior.

## 1. Introduction

The maxillary deficiency in the transverse plane is called maxillary constriction. The main etiologic factors of this deficiency are mouth breathing, harmful habits, like thumb sucking and/or pacifiers, and atypical phonation and swallowing. The poor positioning of the tongue, the imbalance of perioral muscles, the lack of lip seal, together with the labial hypotonicity, contribute to maxillary constriction [1,2,3,4].

Transverse maxillary deficiency is a relatively frequently encountered orthodontic problem, with a prevalence of approximately 10% in adults, and is often characterized by a unilateral or bilateral posterior crossbite [5,6].

The orthodontic procedure used to achieve the correction of maxillary transverse deficiency is called maxillary expansion. The main goal of this treatment is to widen the maxilla by accomplishing the separation of the midpalatal suture (MPS), maximizing skeletal expansion, and minimizing dentoalveolar expansion [7]. This occurs by the stretching of collagenous fibers as well as the local formation of a new bone, correcting the transverse maxillary constriction with a real increase in the transversal width [8].

Currently, there are four expansion treatment modalities used by clinicians: slow maxillary expansion (SME), rapid maxillary expansion (RME), miniscrew assisted rapid palatal expansion (MARPE) and surgically assisted rapid palatal expansion (SARPE). These methods may vary depending on the force used, the appliance, the activation protocol of the screw positioned on the expander and the duration of treatment [9,10]. 

In children and young adolescents, transverse maxillary deficiency is effectively treated with slow or rapid maxillary expansion (RME).

SME typically utilizes continuous low-force systems applied over a longer period of time than RME, and it is achieved through a Quad-helix appliance, removable plates, an spring appliances (e.g., Minne expander) [11,12]. With slow maxillary expansion, tooth movements through the alveolar ridge tend to be greater than the orthopedic effects [13,14]. 

RME is associated with systems of heavy and intermittent forces applied in a short time frame and is achieved through appliances anchored to teeth or tissues (e.g., Hyrax or Haas) in growing patients [11,15,16]. 

In post pubertal patients (young adults and adults), with the closure of the craniofacial sutures and an increase in density of the MPS, RME cannot be performed using the conventional method [17] and surgically assisted techniques (SARPE) are required to provide skeletal expansion [18,19]. In addition to being invasive and expensive, surgically assisted techniques are associated with surgical risks [20,21,22]. 

In this context, as a result of the need to develop a non-surgical treatment for maxillary transverse deficiency in patients who would normally apply for a SARPE, Lee et al. in South Korea and Moon et al. in the USA [23,24] developed a new method named miniscrew-assisted rapid palatal expansion (MARPE).

MARPE is either a tooth-bone borne or a solely bone-borne RPE device with a rigid element that connects to miniscrews inserted into the palate, delivering the expansion force directly to the basal bone of the maxilla [24,25]. 

The time point to shift from a non-surgical to a surgical approach is still not clear enough [26,27]. Existing studies mention that RME should be performed before puberty [28,29] and others have shown successful expansion by RME in adult patients [30,31,32].

Authors have reported histologic studies of patients aged 27, 32, 54, and even 71 [33,34,35,36] years not having signs of MPS fusion. These findings support the existence of great variability in developmental stages in palatal suture fusion and that they are not directly related to chronological age, even more in young adults [33,35,36,37,38]. 

The skeletal maturation assessment is routinely used in the clinical practice of physical and rehabilitation medicine, orthopedics, pediatrics, and orthodontics to plan an adequate treatment in growing subjects [39,40]. 

The gold standard for assessing skeletal maturation is the hand wrist maturation (HWM), a method that needs an extra hand and wrist X-ray [41]. 

In this context, Lamparski et al. [42] introduced a method for assessing cervical vertebral maturation on the cephalometric radiographs. As a result, additional patient radiation was eliminated. Currently, this type of radiograph is routinely applied in orthodontic treatment [43]. 

Cone-beam computed tomography (CBCT) provides 3-dimensional visualization of the MPS in vivo, without any overlapping of anatomic structures, at relatively low cost [17,44]. This could allow the development of a qualitative or quantitative assessment of MPS maturation to assist the decision about whether conventional or surgically assisted maxillary expansion is more appropriate [7]. 

In 2013, Angelieri et al. [27] proposed a new method that allows the individual assessment of MPS using CBCT. They divided MPS maturation into five different stages (A, B, C, D, and E), being able to find the suture open in stages A, B, and C, partially closed in D, and totally closed in E.

Since the creation of this method, some systematic revisions have been performed [45,46,47], but this is the first article to systematically assess and standardize the calibration, training, blinding processes, and randomization of the images that each one of the articles used. This is of vital importance to ensure the internal validity of each study.

Therefore, the present systematic review aimed to critically appraise the available literature on the assessment of maturation of the midpalatal suture before maxillary expansion, using the method proposed by Angelieri et al. [27], with a two-fold focus: (1) to summarize and (2) to assess the methodological quality of evidence.

## 2. Materials and Methods

### 2.1. Protocol and Registration

The systematic review was conducted and written in accordance with the Preferred Reporting Items for Systematic Reviews and Meta-Analyses [48,49]. The study protocol was registered on the International Prospective Register of Systematic Reviews (PROSPERO registration number: CRD42022307742).

### 2.2. Eligibility Criteria

A PICOS (population, intervention, comparator, outcomes, and study design) question was established as an inclusion criteria:

Population (P): human subjects of any gender without restriction of ethnicity or age.

Intervention (I): midpalatal suture maturation method proposed by Angelieri et al. [27] (Table 1).

Condition (C): not having used another method to assess midpalatal suture maturation.

Outcome (O): degree of ossification-maturation-interdigitation of midpalatal suture before maxillary expansion treatment.

Study design (S): observational studies (cohort studies either prospective or retrospective and cross-sectional studies).

Articles including subjects who had undergone any type of orthodontic or orthopedic treatment, nonhuman studies, syndromic conditions, case reports, cleft lip, and palate, and review articles were excluded.

### 2.3. Information Sources and Search Strategy

Electronics searches in MEDLINE (via PubMed), Web of Science, Cochrane Library, Scopus, LILACS and SciELO were conducted up to July 2022. Google Scholar was investigated to partially access the gray literature.

Finally, manual searches in the reference list of included articles were also carried out. There was no restriction of language, year, or status of publication for inclusion.

Detailed search strategies were developed for each database based on the strategy developed for MEDLINE, and subsequently adapted for the other databases (Table 2).

### 2.4. Selection of Sources of Evidence

Study selection was performed in three phases. First, the main researcher (A.S) excluded the duplicate articles using the Reference Manager EndNote X9 (Clarivate Analytics, Philadelphia, Pa). Secondly, two reviewers (A.S and P.S.V) blindly assessed the titles and abstracts of identified records. Then, the same reviewers separately applied eligibility criteria to the full-text studies using the systematic review web application Rayyan [50] (rayyan.qcri.org). Information was cross-checked in a consensus meeting in which disagreements were solved between them. If there was no consensus, a third reviewer was consulted to make a final decision (I.G.C).

### 2.5. Data Charting Process and Data Items

The data was extracted independently by two reviewers (A.S and P.S.V) using a data extraction sheet designed in Microsoft Excel (Redmond, Wash), and any differences were resolved by discussion and consensus with a third reviewer (I.G.C). The following data were extracted from each included study: first author, publication year, study design, sample size, sex distribution, objectives, inclusion criteria, equipment used, number of examiners, calibration, training, and blinding process, inter and intra-evaluator agreement, statistical analysis used, and the author’s conclusion.

### 2.6. Quality Assessment of Included Studies Synthesis of Results

As suggested by Ma et al. [51], the Observational Cohort and Cross-Sectional Studies tool developed by the National Heart, Lung, and Blood Institute [52] was used to assess the quality of the articles that met the inclusion criteria.

Two reviewers independently assessed the articles and subsequently discussed the quality of each study (A.S and P.S.V.). In case of discrepancy, a third author was consulted for further evaluation (I.G.C.).

## 3. Results

A total of 56 studies were identified by electronic searches, and 36 studies remained after removing duplicates. After initial screening, a total of 31 studies met the predetermined inclusion criteria. After the full text review, nine studies were included for this review. In addition, 1 eligible study was identified via hand searches. As a result, 10 studies were included in this systematic review (Figure 1). A summary of the characteristics of each included study is presented in Table 3.

### 3.1. Results of Individual Sources of Evidence and Synthesis of Results

#### 3.1.1. Angelieri et al. 2013

Angelieri et al. [27] assessed midpalatal suture maturation in 140 subjects between the ages of 5–58. Stages A and B were mainly observed up to 13 years of age (55 subjects), whereas stage C was noted primarily from 11 to 17 years of age but occasionally in younger and older age groups (two subjects under 11 years and four over 18 years old). Fusion of the palatine (stage D) and maxillary (stage E) regions of the midpalatal suture was completed after 11 years only in girls (six subjects). From ages 14 to 17 (no years here), three of 13 (23%) boys showed fusion only in the palatine bone (stage D).

#### 3.1.2. Tonello et al. 2017

Tonello et al. [53] evaluated midpalatal suture maturation in 84 children 11–15 years old. Stage A was only found in one 11-year-old girl. In the age group 11–13 years, it was observed that the unfused stages (A, B, and C) were seen in 90.3% of the subjects.

Stage D was present in six girls and five boys (13.1% of the sample). Stage E was found in 10.7% of the sample. Almost all subjects (eight of nine) were 14 or 15 years of age, except for a 12-year-old girl.

#### 3.1.3. Angelieri et al. 2017

Angelieri et al. [54] assessed midpalatal maturation in 78 adults 18–66 years old. Hence, 19 of the adults presented a fused midpalatal suture in the palatine (Stage D) and/or maxillary bones (50, 42 female and eight male). However, the midpalatal suture was not fused in nine of the subjects (12%)

#### 3.1.4. Ladewig et al. 2018

Ladewig et al. [56] evaluated midpalatal suture maturation in 112 patients 16–20 years old. Stage A was found in none of the subjects and Stage B in 16 of them. Stage C was present in 23 males (52.3%) and 27 females (39.7%). Stage D and E were present in 26 and 27 subjects respectively.

#### 3.1.5. Jiménez et al. 2019

Jimenez et al. [57] assessed midpalatal suture maturation in 200 subjects 10–25 years old. They mention that the possibility to find open midpalatal suture (Stages A, B, and C) in individuals 10–15 years old was 70.8% (35 out of 48), in subjects aged 16–20 and 21–25 years old was 21.2% (11 out of 52) and 17% (17 out of 100), respectively. Stage D was present in 58 subjects (nine of them were less than 15 years old) and Stage E was present in 79 subjects (5% of them were less than 15 years old).

#### 3.1.6. Vahdat et al. 2020

Vahdat et al. [58] evaluated MPS maturation in 178 subjects 10–70 years old. Stage A was found in 0 subjects, B in 25 (14 female and 11 male), C in 72 (35 female and 37 male), D in 47 (23 female and 24 male), and E in 34 (17 female and 17 male).

#### 3.1.7. Katti et al. 2020

Katti et al. [59] assessed MPS maturation in 200 subjects between the ages of 11 to 50. The authors didn’t subdivide their results by gender. Stage A was found in 15 subjects (group under 20 years old). Stage B was found in 40 subjects (15 of them older than 20 years old), C in 70 subjects (60 of them older than 20 years old), D in 25, and E in 40 (all of them over 20 years old).

#### 3.1.8. Gatti reis et al. 2020

Gatti Reis et al. [60] evaluated MPS maturation in 487 subjects 15–40 years old. Stage A was found in 0 subjects, B in five (all of them female), C in 166 (93 female and 73 male), D in 81 (38 female and 43 male), and E in 235 (153 female and 82 male).

#### 3.1.9. Villarroel et al. 2021

Villarroel et al. [61] assessed MPS maturation in 150 subjects between the age of 15 and 30 years old. Stage A was found in 0 subjects, B in two (one female and one male), C in 65 (29 female and 36 male), D in 33 (17 female and 16 male), and E in 50 (30 female and 20 male).

### 3.2. Quality Assessment of Included Studies

The obtained grade of quality assessment for each study is included in Table 4. Grades for the selected studies ranged from 58.3% to 83.3%. One study [59] had poor quality, seven studies [27,53,54,55,56,58,60] had fair quality, and two studies [57,61] had good quality.

## 4. Discussion

### 4.1. Summary of Evidence

One of the most important factors when making the clinical decision regarding how to deal with a transverse maxillary constriction is defining whether the midpalatal suture is open or closed, thus influencing enormously the treatment that will be given to the pa-tient. This can be especially challenging in late-stage adolescent and young adult patients because there is no consensus in the literature regarding the minimum age for reliable palatal expansion [57]. 

Even though RME is a more conservative treatment, if it is indicated in patients with totally or partially closed midpalatal sutures, it can lead to consequences such as significant pain, gingival recession, palatal mucosa ulceration or necrosis, buccal tipping of the posterior teeth, reduction of buccal bone thickness [30,62,63,64,65,66], alveolar bone bending [67], buccal root resorption [68], fenestration of the buccal cortex [69], and instability of the expansion [70,71]. On the other hand, it is important to mention that even though a surgical expansion with SARPE is possible at any time throughout life, it implies increasing morbidity, cost, risk, and more days required for patient recovery [54]. It has also been reported to be the most unpredictable procedure among all orthognathic surgery modal-ities. This unpredictability of the surgical expansion has to do with its relapse potential [72,73]. 

A third option mentioned in the scientific literature is the use of micro implants (MARPE) in cases in which the midpalatal suture is in process of closure [74,75,76]. 

Despite the unquestionable success of the RME protocol in clinical practice, there is still no consensus regarding the age limit for palatal expansion. This is mainly due to the great physiologic variability, among patients with an obliterated palatal suture earlier or at a more advanced age, without a precise diagnostic [77]. This has been confirmed by histologic studies that have shown the same variability in the maturation of the midpalatal suture [35,36,37,38]. 

Furthermore, some clinicians have recommended surgical intervention in patients older than 14 years [78], 16 years [79], 20 years [80], or 25 years [81]. To add to the confusion, many case reports have shown that RME is possible in older adult patients [30,32,62]. 

As mentioned above, a lot of uncertainty and doubt exists in scientific literature because of contradictory information in relation to which is the best clinical approach when performing maxillary expansion.

Within this frame of reference, a diagnostic method in which it is possible to evaluate the maturation of the palatal suture with safety and reliability before maxillary expansion becomes important [53]. 

The individual evaluation of midpalatal suture maturation on CBCT scans has been proposed by Angelieri et al. [27], to identify the morphology of the midpalatal suture prior to maxillary expansion, trying to guide clinicians in choosing the best clinical procedure to accomplish a successful treatment.

Two important factors mentioned in literature are age and sex. They play an essential role in finding midpalatal suture opening but are not crucial in the decision making because they are not reliable parameters to determine if the MPS is merged or not [35]. Angelieri et al. [27] mention that chronologic age is unreliable for determining the developmental status of the suture during growth, even though it has been suggested that gradual obliteration of MPS occurs as patients get older.

In addition, it has been mentioned that skeletal maturation of the MPS occurs earlier in women than in men [82,83,84,85], especially in the puberal ages [86,87]. This is also possible to observe in all the studies included in this revision [27,53,54,56,57,58,59,60,61]. 

Related to age, an interesting event that occurs is related to studies that included adults. In them, it is possible to appreciate subjects that, despite having passed their growth phase, still have an open or partially obliterated midpalatal suture [27,53,54,56,57,58,59,60,61] (Table 5). 

Besides age and sex, it is important to consider the existence of other biological factors responsible for the resistance to maxillary transverse expansion, other than the stage of MPS maturity, such as bone density [7,33,88,89], fusion of the zygomaticotemporal, zygomaticofrontal, zygomaticomaxillary, and pterygopalatine sutures [33,36,69,90,91,92,93]. 

Even though the advent of CBCT has brought many benefits to the field of orthodontics, allowing the clinician to three-dimensionally visualize the maxillary anatomy [94] and evaluate the MPS maturation without the overlap of the surrounding structures [38], we have to remember that radiological assessment is not a risk-free procedure, especially when children are involved, and there is a growing concern of radiation dose [95,96].

The existing guidelines about the use of CBCT in orthodontics have emphasized the need of a stronger justification when prescribing CBCT examinations. Children or young adults should undergo a CBCT examination only when the benefits of the diagnosis or treatment plan outweigh the potential risks of radiation exposure [97]. Jimenez et al. [57] mention that the need for a tomographic examination should be reduced to avoid the load of ionizing radiation in the patient. In this regard, it is an essential clinical procedure to follow the guidelines of imaging proposed by the American Academy of Oral and Maxillofacial Radiology appropriately [97], according to the clinical condition and assessing the radiation dose risk.

When assessing the reliability and reproducibility of the method proposed by Angelieri et al. [27], we were able to find contradictory information. Some authors mentioned that the method presents a substantial reliability and reproducibility as evaluated through the intraexaminer and interexaminer reliability calculation [27,52], while other authors emphasized the low reproducibility of the method [47,98], describing it as non-intuitive and requiring major training for operator calibration.

Vieira Barbosa et al. [55] mention that whenever proposing a new diagnostic method examiner’s agreement plays an important role. Methods that are considered highly reproducible are also considered reliable. Reliability is the capacity of a method to result in identical or similar outcomes in different clinical or statistical experiments. More specifically, any test or procedure considered reliable will always result in similar outcomes regardless of the time, environment, or examiner. This reliability helps reduce the occurrence of diagnostic errors.

Apart from reliability and reproducibility of a method, validation of a diagnostic method is also necessary, with this method being validated in the literature [77]. Specifically, the individual assessment of midpalatal suture maturation was compared with hand–wrist and cervical vertebrae maturation and showed strong statistical association.

### 4.2. Methodological Quality Assessment

Methodological quality (risk of bias) assessment is an important step before study initiation usage. Therefore, accurately judging study type is the first priority, and choosing the proper tool is also important [51]. One of the strengths of this review is that it is the first to assess the methodological quality of the articles related to this topic using the Quality assessment of the included studies using the Observational Cohort and Cross-Sectional Studies tool.

This differentiates this systematic review from another [26] in which a quality review was performed using the STROBE checklist. According to Ma et al. [51], this is not the most suitable tool for quality assessment of cross-sectional studies.

Of the seven studies included, only two [57,61] described and defined clearly the study population, mentioning in detail the demographic background, location, and time period for obtaining the samples. Seven out of 10 studies [54,55,56,57,58,60,61] mentioned how the calculations of their sample size was done. Similarly, seven out of 10 [27,53,54,55,56,57,61] studies kept the examiners blinded (Table 4).

Other points of vital importance are related to the calibration between the observers, the blinding process, and the randomization of the images used to evaluate the intra-observer agreement.

Only four studies carried out a calibration process and prior training [27,56,57,61]. Only five studies randomized the images of the second examination [27,54,55,56,57,60] and five included all the images for the second examination [53,55,56,57,59] (Table 6).

### 4.3. Limitations

A limitation of this study has to do with results not being homogeneous, making it impossible to perform a meta-analysis.

The methodological quality of the studies included was assessed rigorously and many deficiencies were found, such as: lack of randomization, blinding, and sample calculation.

Another limitation has to do with the method itself. Thus, because of its qualitative nature, an extensive calibration and training program is necessary for more reliable and reproducible applications [55]. 

There is an urgent need for future studies to also include the evaluation of the rest of the circummaxillary sutures.

## 5. Conclusions

The midpalatal suture maturation method has the potential to be used for diagnostic purposes.

Before using this method, an extensive training and calibration program should be performed.

Even though age and sex play an important role in midpalatal suture obliteration, every patient should be assessed individually before choosing the best protocol for maxillary expansion.

## Figures and Tables

**Figure 1 diagnostics-12-02774-f001:**
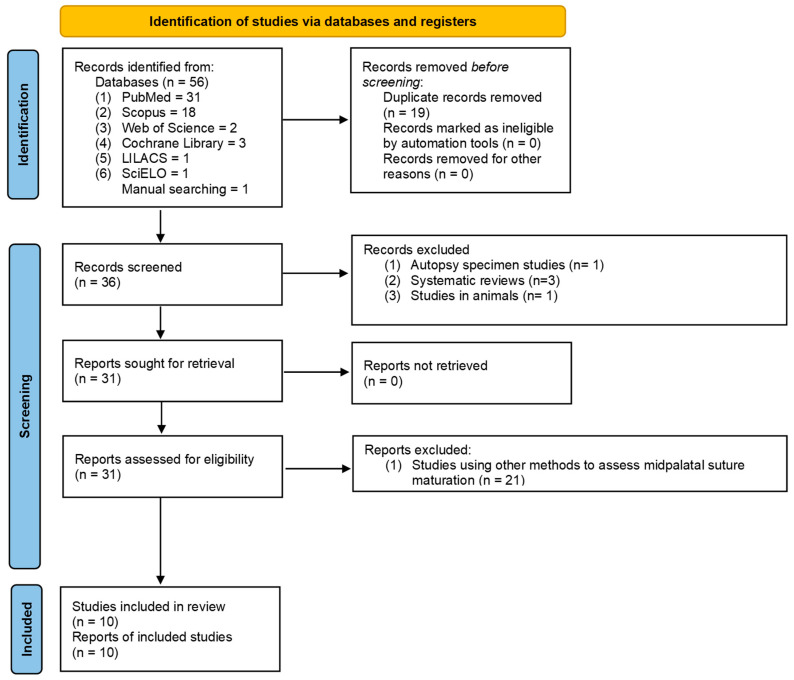
PRISMA 2020 flow diagram for new systematic reviews which included searches of databases and registers only. Page, MJ.; McKenzie, JE.; Bossuyt, PM.; Boutron, I.; Hoffmann, TC.; Mulrow, CD.; et al. The PRISMA 2020 statement: an updated guideline for reporting systematic reviews. BMJ 2021;372: n71. doi: 10.1136/bmj.n71 [48].

**Table 1 diagnostics-12-02774-t001:** Skeletal maturation stages of the MPS proposed by Angelieri et al. [27].

Stage	Description
A	Represents the earliest maturation stage of the suture, and in this stage the suture was identified as a relatively straight high-density line at the midline
B	The suture presents an irregular shape and was identified as a scalloped high-density line at the midline
C	The suture is seen as two parallel, scalloped, high-density lines close to each other and separated in some areas by small low-density spaces
D	The complete fusion of the suture has occurred in the palatine bone and the radiographic image of the suture was identified as two scalloped, high-density lines at the midline on the maxillary portion of the palate that were not visible in the palatine bone
E	Fusion of the suture has occurred in the maxilla. It is not possible to identify the MPS. As to bone density, it is the same as in other parts of the palate

**Table 2 diagnostics-12-02774-t002:** Electronic literature search strategy.

Database	Keywords	Time Frame	Result	Included Articles
MEDLINE/PubMed	(“midpalatal suture maturation” OR “midpalatal suture maturation method”) AND (“cranial suture” OR “cranial sutures” OR “midpalatal suture”) AND (“maturation” OR “interdigitation” OR “ossification”) AND (“evaluation” OR “assess” OR “assessment”)	January 2013–July 2022	31	10
Scopus			18	
Web of Science			23	
Cochrane Library			3	
LILACS			1	
SciELO			1	

**Table 3 diagnostics-12-02774-t003:** Summary of characteristics of included studies.

Authors (y)	Country	Sample Size	Age	Study Design	Equipment Used	Specifications
Angelieri et al. (2013) [27]	United States	140 (54 M–86 F)	5.6–58.4 years	Cross-sectional	iCAT cone-beam 3D imaging system	8.9 to 20 s, FoV at least 11 cm, voxel 0.2 a 0.3 mm
Tonello et al. (2017) [53]	Brazil	84 (40 M–44 F)	11–15 years	Cross-sectional	iCAT scanner	8.9 to 30 s, FoV at least 11 cm, voxel 0.2 a 0.3 mm
Angelieri et al. (2017) [54]	Brazil	78 (14 M–64 M)	18–66 years	Cross-sectional	iCAT Cone Beam 3D Imaging system scanner	17.8 s, coxel 0.3 mm
Barbosa et al. (2018) [55]	Brazil	60 (27 M–33 F)	11–21 years	Cross-sectional	Not mentioned	Not mentioned
Ladewig et al. (2018) [56]	Brazil	112 (68 M–44F)	16–20 years	Cross-sectional	iCAT scanner	40 s, FoV 22 × 16 cm, 120 kV, 36 mA, 0.4 voxel
Jimenez et al. (2019) [57]	Peru	200 (95 M–105 F)	10–25 years	Cross-sectional	Planmeca ProMax 3D Mid scanner	13.68 s, FoV at least 11 cm,90 kV, 10 mA, 0.2 a 0.3 mm voxel
Vahdat et al. (2020) [58]	Iran	178 (89 M–89 F)	10–70 years	Cross-sectional	Newtom VGi Cone Beam CT	18 s, 110 kV, 1–20 mA.
						The equipment was automatically adjusted
Katti et al. (2020) [59]	India	200 (95 M–105 F)	11–50 years	Cross-sectional	NewTom Giano CBCT machine	Not mentioned
Gatti Reis et al. (2020) [60]	Brazil	487 (198 M–289 F)	15–40 years	Cross-sectional	iCAT scanner	120 kV, 8 mA, 26.9 s rotation, 0.25 mm voxel
						FOV between 6 × 23 and 8 × 23 cm.
Villarroel et al. (2021) [61]	Chile	150 (73 M–77 F)	15–30 years	Cross-sectional	Sirona Ortophos XG3D	14 segundos, FoV 8 × 8, 85 kV, 7 mA, voxel 0.16

F, female; M, male; Y, years; FoV, Field of View.

**Table 4 diagnostics-12-02774-t004:** Blinding, Calibration process of included studies.

Authors (y)	Nº Examiners	Calibration-Validation Process	Intraexaminer Agreement	Interexaminer Agreement	Washout Period	Images Included in Second Examination	Randomization of Images (Second Examination)	Blinding
Angelieri et al. (2013) [27]	3	Yes (10 images calibration–30 images calibration)	*K*: 0.77 (0.75–0.79)	*K*: 0.87 (0.82–0.93)	2 days	30 images	Yes	Yes
Tonello et al. 2017 [53]	2	Not mentioned	Not mentioned	Not mentioned	15 days	All images	Not mentioned	Yes
Angelieri et al. 2017 [54]	1	Not mentioned	*K*: 0.80	Not applicable	30 days	30 images	Yes	Yes
Barbosa et al. 2018 [55]	21	Not mentioned	*K*: 0.42	*K*: 0.34	21 days	All images	Yes	Yes
Ladewig et al. 2018 [56]	2	Yes (used images included in main study)	*K*: 0.87	*K*: 0.89	15 days	All images	Not mentioned	Yes
Jimenez et al. (2017) [57]	2	Yes (not clear if they used same images included in main study)	*K*: 0.89	*K*: 0.90	30 days	All images	Yes	Yes
Vahdat et al. (2018) [58]	1	Not mentioned	Not mentioned	Not applicable	Not mentioned	Not mentioned	Not mentioned	Not mentioned
Katti et al. 2020 [59]	1	Not mentioned	ICC > 0.8	Not applicable	5 days	All images	Not mentioned	Not mentioned
Gatti Reis et al. 2020 [60]	1	Not mentioned	*K:* 0.8774	Not applicable	30 days	49 images	Yes	Not mentioned
Villarroel et al 2021 [61]	1	Yes (used 10 images)	PCC: 0.94	PCC 1.0	Not mentioned	Not mentioned	Not mentioned	Yes

K: Cohen’s kappa; ICC: interclass correlation; PCC: Pearson correlation coefficient.

**Table 5 diagnostics-12-02774-t005:** Quality assessment of the included studies using the Observational Cohort and Cross-Sectional Studies (NHBLI) tool.

Included Studies	Quality Assessment Criteria														Quality Score (%)
	1	2	3	4	5	6	7	8	9	10	11	12	13	14	
Angelieri et al. 2013 [27]	Yes	No	Yes	Yes	No	No	No	Yes	Yes	NA	Yes	Yes	NA	Yes	8/12 (66.6%)
Tonello et al. 2017 [53]	Yes	No	Yes	Yes	No	No	No	Yes	Yes	NA	Yes	Yes	NA	Yes	8/12 (66.6%)
Angelieri et al. 2017 [54]	Yes	No	Yes	Yes	Yes	No	No	Yes	Yes	NA	Yes	Yes	NA	Yes	9/12 (75%)
Barbosa et al. 2018 [55]	Yes	No	Yes	Yes	Yes	No	No	Yes	Yes	NA	Yes	Yes	NA	Yes	9/12 (75%)
Ladewig et al. 2018 [56]	Yes	No	Yes	Yes	Yes	No	No	Yes	Yes	NA	Yes	Yes	NA	Yes	9/12 (75%)
Jimenez et al. 2019 [57]	Yes	Yes	Yes	Yes	Yes	No	No	Yes	Yes	NA	Yes	Yes	NA	Yes	10/12 (83.3%)
Vahdat et al. 2020 [58]	Yes	No	Yes	Yes	Yes	No	No	Yes	Yes	NA	Yes	No	NA	Yes	8/12 (66.6%)
Katti et al. 2020 [59]	Yes	No	Yes	Yes	No	No	No	Yes	Yes	NA	Yes	No	NA	Yes	7/12 (58.3%)
Gatti Reis et al. 2020 [60]	Yes	No	Yes	Yes	Yes	No	No	Yes	Yes	NA	Yes	No	NA	Yes	8/12 (66.6%)
Villarroel et al. 2021 [61]	Yes	Yes	Yes	Yes	Yes	No	No	Yes	Yes	NA	Yes	Yes	NA	Yes	10/12 (83.3%)

CD: cannot determine; NA: not applicable; NR: not reported; NHBLI: National Heart, Blood and Lung Institute, United States. (1) Was the research question or objective in this paper clearly stated? (2) Was the study population clearly specified and defined? (3) Was the participation rate of eligible persons at least 50 %? (4) Were all the subjects selected or recruited from the same or similar populations? Were inclusion and exclusion criteria for being in the study prespecified and applied uniformly to all participants? (5) Was a sample size justification, power description or variance and effect estimates provided? (6) For the analyses in this paper, were the exposure(s) of interest measured prior to the outcome(s) being measured? (7) Was the timeframe sufficient so that one could reasonably expect to see an association between exposure and outcome if it existed? (8) For exposures that can vary in amount or level. did the study examine different levels of the exposure as related to the outcome (e.g., categories of exposure or exposure measured as continuous variable)? (9) Were the exposure measures (independent variables) clearly defined, valid, reliable, and implemented consistently across all study participants? (10) Was the exposure(s) assessed more than once over time? (11) Were the outcome measures (dependent variables) clearly defined, valid, reliable, and implemented consistently across all study participants? (12) Were the outcome assessors blinded to the exposure status of participants? (13) Was loss to follow-up after baseline 20 % or less? (14) Were key potential confounding variables measured and adjusted statistically for their impact on the relationship between exposure(s) and outcome(s)?

**Table 6 diagnostics-12-02774-t006:** Distribution of maturational stages of midpalatal suture by age in the included studies.

Angelieri et al. 2013 [27]	Age groups	**5 < 11 y**	**11–<14 y**	**14–18 y**	**>18 y**		
	MPS Stages (n)	**A**:4; **B**:22; **C**:2; **D**:0; **E**:0	**A**:1; **B**:28; **C**:13; **D**:1; **E**:5	**A**:0; **B**:6; **C**:12; **D**:6; **E**:8	**A**:0; **B**:1; **C**:4; **D**:10; **E**:17	-	-
Tonello et al. 2017 [53]	Age groups	**11**	**12**	**13**	**14**	**15**	
	MPS Stages (n)	**A**:1; **B**:4; **C**:8; **D**:0; **E**:0	**A**:0; **B**:9; **C**:14; **D**:3; **E**:1	**A**:0; **B**:5; **C**:6; **D**:1; **E**:0	**A**:0; **B**:1; **C**:8; **D**:3; **E**:3	**A**:0; **B**:2; **C**:2; **D**:3; **E**:4	-
Angelieri et al. 2017 [54]	Age groups	**<30 y**	**>30 y**				
	MPS Stages (n)	**A**:0; **B**:1; **C**:3; **D**:11; **E**:21	**A**:0; **B**:2; **C**:3; **D**:8; **E**:29	-	-	-	-
Ladewig et al. 2018 [56]	Age groups	**16**	**17**	**18**	**19**	**20**	
	MPS Stages (n)	**A**:1; **B**:3; **C**:9; **D**:4; **E**:5	**A**:0; **B**:4; **C**:13; **D**:5; **E**:4	**A**:0; **B**:0; **C**:13; **D**:5; **E**:6	**A**:0; **B**:0; **C**:7; **D**:8; **E**:8	**A**:0; **B**:1; **C**:8; **D**:8; **E**:4	
Jimenez et al. 2019 [57]	Age groups	**10–15**	**16–20**	**21–25**			
	MPS Stages (n)	**A**:2; **B**:13; **C**:20; **D**:9; **E**:4	**A**:0; **B**:1; **C**:10; **D**:21; **E**:20	**A**:0; **B**:2; **C**:15; **D**:28; **E**:55	-	-	-
Katti et al. 2020 [58]	Age groups	**11–20**	**21–30**	**31–40**	**41–50**		
	MPS Stages (n)	**A**:15; **B**:25; **C**:10; **D**:0; **E**:0	**A**:0; **B**:5; **C**:30; **D**:0; **E**:0	**A**:0; **B**:5; **C**:20; **D**:15; **E**:10	**A**:0; **B**:5; **C**:10; **D**:5; **E**:25	-	-
Vahdat et al. 2020 [59]	Age groups	**10–19**	**20–29**	**30–39**	**40–49**	**50–59**	**60–69**
	MPS Stages (n)	**A**:0; **B**:12; **C**:0; **D**:0; **E**:0	**A**:0; **B**:6; **C**:7; **D**:5; **E**:0	**A**:0; **B**:2; **C**:26; **D**:12; **E**:2	**A**:0; **B**:0; **C**:20; **D**:13; **E**:7	**A**:0; **B**:1; **C**:11; **D**:8; **E**:17	**A**:0; **B**:3; **C**:5; **D**:5; **E**:7
Gatti Reis et al. 2020 [60]	Age groups	**15–20**	**21–25**	**26–30**	**31–35**	**36–40**	
	MPS Stages (n)	**A**:0; **B**:0; **C**:43; **D**:17; **E**:34	**A**:0; **B**:1; **C**:71; **D**:34; **E**:102	**A**:0; **B**:1; **C**:25; **D**:15; **E**:44	**A**:0; **B**:2; **C**:13; **D**:10; **E**:32	**A**:0; **B**:1; **C**:14; **D**:5; **E**:23	**-**
Villaroel et al. 2021 [61]	Age groups	**15–20**	**21–25**	**26–30**			
	MPS Stages (n)	**A**:0; **B**:0; **C**:32; **D**:10; **E**:7	**A**:0; **B**:2; **C**:18; **D**:8; **E**:23	**A**:0; **B**:0; **C**:15; **D**:20; **E**:20	-	-	-

## Data Availability

Not applicable.

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
