# Peer review of "Midpalatal Suture Maturation Method for the Assessment of Maturation before Maxillary Expansion: A Systematic Review"

_diagnostics, 2022, doi:10.3390/diagnostics12112774_

Round 1

Reviewer 1 Report

Manuscript ID: diagnostics-1979821

Midpalatal suture maturation method for the assessment of maturation before maxillary expansion: a systematic review

This manuscript analyzes the midpalatal suture maturation method used for diagnostic. I think it's very interesting and has clinical value. However, there are several suggestions that can hopefully improve.

1. Please reinforce the difference between Rapid maxillary expansion (RME) and slow maxillary expansion. In addition, it can be explained what other methods are currently in clinical use.

2. It is recommended that additional diagrams be added to illustrate RME, MARPE, SARME and MPS.

3. The aim of this study was to systematically identify, evaluate and provide a synthesis of the existing literature about this new method and to rigorously assess the methodological quality of these articles.

The purpose of this manuscript is too vague and could be more focused.

4. Conclusions should state when the various devices were used.

5. Angelieri et al. 2013 It has been nine years since today.

Does the relationship of manuscript age affect the results? After all, the evolution of dentistry is quite rapid.

Author Response

Dear reviewer,

We have answered all your corrections in the attached pdf. Thank you very much for your time and help.

Best regards.

Reviewer 2 Report

Dear Authors,

Assessment of midpalatal suture maturation is crucial before deciding which type of maxillary expansion technique will be performed to treat transverse discrepancies. In 2013, Angelieri et al. proposed a new method to evaluate midpalatal maturation using cone-beam computed tomography. The aim of this study was to systematically identify, evaluate and provide a synthesis of the existing literature about this new method and to rigorously assess the methodological quality of these articles.

The study is of scientific interest and in line with the aims of the journal. However, there are some issues that should be added.

Abstract

  • Abstract: The abstract should be a total of about 200 words maximum. The abstract should be a single paragraph and should follow the style of structured abstracts, but without headings: 1) Background: Place the question addressed in a broad context and highlight the purpose of the study; 2) Methods: Describe briefly the main methods or treatments applied. Include any relevant preregistration numbers, and species and strains of any animals used. 3) Results: Summarize the article's main findings; and 4) Conclusion: Indicate the main conclusions or interpretations. The abstract should be an objective representation of the article: it must not contain results which are not presented and substantiated in the main text and should not exaggerate the main conclusions. 

(https://www.mdpi.com/journal/diagnostics/instructions). 

Please remove headings. 

Introduction

-       I suggest revising the Introduction section. In my opinion, at the beginning of the Introduction section, the authors should define the transverse maxillary constriction reporting definition, epidemiology, etiological factors, and common treatments. 

Moreover, I suggest better clarifying the gap existing in the scientific literature and the rationale.

-       Moreover, after lines 55-56, in my opinion it was very important to report the most widely used methods to assess skeletal maturation, as the hand and wrist rx as gold standard, and the CVM by Baccetti. Two systematic reviews were conducted on this topic in the last years, and authors concluded that the CVM method shows a high level of correlation with the HWM method. Please refer to and cite: Szemraj et al. Is the cervical vertebral maturation (CVM) method effective enough to replace the hand-wrist maturation (HWM) method in determining skeletal maturation?-A systematic review. Eur J Radiol. 2018 May;102:125-128. doi: 10.1016/j.ejrad.2018.03.012. and Ferrillo et al. Reliability of cervical vertebral maturation compared to hand-wrist for skeletal maturation assessment in growing subjects: A systematic review. J Back Musculoskelet Rehabil. 2021;34(6):925-936. doi: 10.3233/BMR-210003.

-       I suggest modify the references 9 and 10 with the most recent literature: Ventura V, Botelho J, Machado V, Mascarenhas P, Pereira FD, Mendes JJ, Delgado AS, Pereira PM. Miniscrew-Assisted Rapid Palatal Expansion (MARPE): An Umbrella Review. J Clin Med. 2022 Feb 26;11(5):1287. doi: 10.3390/jcm11051287. Suri L, Taneja P. Surgically assisted rapid palatal expansion: a literature review. Am J Orthod Dentofacial Orthop. 2008 Feb;133(2):290-302. doi: 10.1016/j.ajodo.2007.01.021. Lin JH, Li C, Wong H, Chamberland S, Le AD, Chung CH. Asymmetric Maxillary Expansion Introduced by Surgically Assisted Rapid Palatal Expansion: A Systematic Review. J Oral Maxillofac Surg. 2022 Aug 19:S0278-2391(22)00782-0. doi: 10.1016/j.joms.2022.08.008. 

Material and methods

-       I suggest to modify “The study protocol was registered on the International Prospective Register of Systematic Reviews (CRD42022307742)” to “The study protocol was registered on the International Prospective Register of Systematic Reviews (PROSPERO registration number: CRD42022307742)”

References 

References should be written according to the Instruction for Authors:

Journal Articles:
1. Author 1, A.B.; Author 2, C.D. Title of the article. Abbreviated Journal Name YearVolume, page range.

 (https://www.mdpi.com/journal/diagnostics/instructions)

Finally, I would like to congratulate with authors for the interesting research they conducted. 

Author Response

Dear reviewer,

We have answered all your corrections in the attached pdf. Thank you very much for your time and help in improving our article.

Best regards.

Round 2

Reviewer 2 Report

Dear authors,

In my opinion the paper is ready for the publication.

I found this work impactful and fit well with in the scope of this journal.